# A Data-Driven DAE-CNN-BiLSTM-Attention Prediction Model for the State of Health of Lithium-ion Batteries

1st Can Zhang
*School of Electrical Engineering*
*Southwest Jiaotong University*
Chengdu, China
zhangcan@my.swjtu.edu.cn

2nd Yuanjiang Hu
*School of Electrical Engineering*
*Southwest Jiaotong University*
Chengdu, China
360832800@qq.com

3rd Deqing Huang
*School of Electrical Engineering*
*Southwest Jiaotong University*
Chengdu, China
elehd@home.swjtu.edu.cn

4th Jiaxin Fang
*School of Electrical Engineering*
*Southwest Jiaotong University*
Chengdu, China
2022210644@my.swjtu.edu.cn

5th Na Qin
*School of Electrical Engineering*
*Southwest Jiaotong University*
Chengdu, China
qinna@swjtu.edu.cn

*Abstract*—Accurately predicting the health state of lithium-ion batteries is essential for their safety, reliability, and longevity. Predicting State of Health (SOH) using health indicators is a proven and effective method. However, real-world battery charge-discharge data is often noisy, particularly during capacity regeneration. To achieve accurate health state predictions, we extracted over ten health indicators and designed a hybrid model: DAE-CNN-BiLSTM-Attention. This model integrates the strengths of Convolutional Neural Networks (CNN) for local feature extraction, Bidirectional Long Short-Term Memory networks (BiLSTM) for temporal dependency learning, the Attention mechanism for effective weight assignment, and Denoising Autoencoders (DAE) for restoring original data, enabling the network to better adapt to complex real-world environments. The adaptability and stability of the proposed model were validated using two public datasets: NASA and CALCE. Compared to other existing methods, the proposed model demonstrated superior performance, achieving mean absolute error (MAE) and root mean square error (RMSE) of 0.0154 and 0.0191, respectively.

*Index Terms*—State of Health(SOH), Lithium-ion Battery(LiB), convolution neural network(CNN), feature extraction, long short-term memory (LSTM)

## I. INTRODUCTION

To address environmental challenges and the fossil energy crisis, there is an urgent and vigorous development of clean energy sources such as hydro, wind, and nuclear power. Consequently, the issue of energy storage and utilization has become particularly critical. Lithium batteries, compared with other types of batteries, such as NiMH batteries and lead-acid batteries, offer higher energy density, lower self-discharge rates, and longer charge-discharge lifespans. These advantages have led to their widespread application, including in electric vehicles, portable electronics, and energy storage systems [1]. However, over time and with usage, batteries inevitably experience aging. This results in increased internal resistance, reduced usable capacity, and degraded performance, which can lead to battery leakage, localized short circuits, and potential safety hazards such as device malfunctions, shutdowns, or even overheating and explosions. Consequently, in critical applications, batteries are often replaced periodically to ensure safety, which inevitably leads to resource wastage. Battery Management Systems (BMS) are essential to ensure that batteries function safely, reliably, and efficiently, with the health state being a core concern. Accurately predicting the SOH is vital for assessing battery aging, conserving resources, and ensuring battery safety [2]- [4].

The health status of the battery (SOH) is a key indicator of its performance deterioration, quantifying the rate of battery aging by a percentage. As batteries age, the percentage gradually decreases, a phenomenon commonly described as a reduction in the total available capacity of the battery and an increase in resistance. The SOH value directly reflects the current health of the battery; the higher the value, the better the battery state. To accurately estimate the SOH of a battery, researchers have developed various monitoring technologies to monitor the voltage, current, and temperature of the battery in real-time. In general, battery health is measured by the ratio of the maximum available capacity to the rated capacity of the battery [5]. This ratio can be used to predict the health of the battery and to advise users on when to replace it. Therefore, understanding the SOH of the battery is essential to extending its life and ensuring safe operation. Through accurate monitoring and analysis of these parameters, manufacturers can adjust the strategy of the battery management system to maintain optimal performance and prolong the life of the battery. This paper also adopts this definition, with the SOH defined as shown in Eq (1).

$$SOH = \frac{C_{\text{max}}}{C_{\text{norm}}} \times 100\% \qquad (1)$$

where $C_{\text{max}}$ and $C_{\text{norm}}$ are two key parameters, representing the actual maximum capacity and the standard rated capacity of the battery, respectively. In the field of battery data-driven research, if the maximum available capacity of the battery falls below 70% of its initial value, it is usually regarded as a warning line, known as a failure threshold. Such a situation in high-speed rail (HSR) and electric vehicle (EV) batteries could indicate serious aging or health problems that require timely attention and maintenance. According to the relevant literature [6], the battery in this case should not be used for high-load or long-duration applications to avoid potential safety risks.

Due to the complex operating environments of batteries, such as temperature variations and the internal chemical reactions within the battery, which introduce uncertainties, the time-varying and highly nonlinear characteristics of batteries make accurately predicting SOH a challenging research problem [7]. Currently, these technologies can be divided into two main categories: model-based and data-driven. Battery fault diagnosis technology based on the model method predicts the health of the system by extracting model parameters. Through in-depth study and detailed analysis of the physical and chemical properties of the battery, the equivalent circuit model is constructed to accurately simulate the behavior of the battery [8]- [9] or electrochemical models [10]. Typically, state observers are used to describe the degradation mechanisms between battery cycles [11], such as Kalman filters [12]- [13] and particle filters [15]. Although electrochemical models have relatively high accuracy, they rely on precise electrochemical impedance spectroscopy. On the other hand, equivalent circuit models are less satisfactory because they fail to capture the aging characteristics of the battery. Model-based approaches often involve ideal or empirical models that do not account for internal chemical reactions and aging mechanisms, making accuracy increasingly difficult to maintain over time [1]. Additionally, the physical and chemical parameter models of batteries are very complex, which imposes severe limitations due to measurement difficulties, robustness, dynamic accuracy, and poor adaptability.

In contrast to the model-based approach, the data-driven approach does not require consideration of complex physical and chemical parameters. Instead, it directly extracts and analyzes historical charge and discharge data from the battery. By using machine learning or deep learning techniques to delve into the rich information hidden in the data, the relevance of these characteristics to the state of health (SOH) is revealed. Examples include Support Vector Machines (SVM) [17], Backpropagation (BP) neural networks [16], Relevance Vector Machines (RVM) [18], and Bayesian networks [19]. However, considering the time dependency of battery degradation data, recurrent neural networks (RNNs) have shown superior predictive performance. Literature [11] has already suggested using RNNs for battery SOH prediction. LSTM, as an upgraded version of RNNs [20], prevents issues like gradient explosion

and performs exceptionally well in sequence prediction. To connect the degradation data over time, some studies have used bidirectional LSTM networks [21].In order to overcome the limitations of single network models, many researchers use hybrid network technology to improve prediction performance. For example, in [22], the LSTM network is used to predict battery life using the empirical mode decomposition (EMD) method. This decomposition method can capture the complex dependencies between different states within the battery, thus providing a more accurate model for battery life prediction. The article [23] demonstrates how to combine a gated recurrent unit (GRU) with a convolutional neural network (CNN) to predict the state of health (SOH) of lithium-ion batteries. By combining the advantages of the two neural networks, this study aims to gain a more comprehensive understanding of the degradation process of battery performance and to assess its remaining useful life (RUL). The study [24] estimates SOH and predicts RUL by constructing a hybrid network of LSTM and CNN (CNN-LSTM). This hybrid network combines the ability of two neural networks to process sequence data and image data, leveraging their respective strengths to achieve higher prediction accuracy and efficiency.

In exploring the estimation of lithium-ion battery life (SOH), researchers not only limited themselves to using different algorithms but also examined a range of health indicators (HIs). These include the constant current constant voltage scheme [25], Open Circuit Voltage (OCV) [26], Incremental Capacity (IC) curve peaks [27], cycle numbers [28], differential capacity [29], and differential voltage [30], all of which describe battery degradation. External characteristic parameters such as current, voltage, and temperature are used as health indicators [31], and the HIs closely related to attenuation are screened through Pearson correlation analysis.

Despite achieving good prediction results, most existing studies are based on neural network (NN) training where the hidden layer features are weighted equally across dimensions. However, each feature has a different effect on the SOH, and ignoring this factor can affect prediction accuracy. Attention mechanisms, including channel attention (dimension attention), multiple attention [11], spatial attention, and temporal attention [32], improve the performance of the network model by dynamically focusing on the key information related to tasks. These mechanisms can identify data points that are of greater importance in a particular context, thereby enhancing the model's ability to understand and predict complex scenarios. Moreover, real-world data are often noisy, especially during the capacity regeneration process [33].

In this paper,considering the impact of more than 10 health factors and noise on battery aging, we propose a novel hybrid network model designed for accurate State of Health (SOH) prediction of batteries. The proposed model combines advanced neural network architectures with a focus on feature extraction and sequence analysis. It follows a multi-step process, beginning with data denoising to reduce the impact of noise and improve the quality of the input data. Subsequently, local features are extracted using two convolutional layers,

which effectively capture spatial patterns in the data. These layers are crucial for identifying significant local variations that contribute to battery aging.

The model then employs a Bidirectional Long Short-Term Memory (BiLSTM) network to capture long-term dependencies within the sequence data. This step is essential for understanding the temporal dynamics of battery degradation, as it allows the model to learn from both past and future states of the sequence. To further enhance the handling of sequential data, a temporal attention mechanism is integrated into the model. This mechanism assigns weights to each timestep, enabling the network to focus on the most critical moments within the sequence, thereby improving prediction accuracy.

The primary contributions of this paper are as follows:

1. We validate the accuracy and feasibility of the proposed DAE-CNN-BiLSTM-Attention model for SOH prediction using two widely recognized public datasets, NASA and CALCE. The results demonstrate that our model outperforms existing methods in terms of Mean Absolute Error (MAE) and Root Mean Square Error (RMSE), confirming its effectiveness in practical applications.

2. We consider the impact of more than ten health indicators on battery aging, including factors such as time, temperature, voltage, current, and internal resistance. Additionally, we account for the influence of noise on the data. To minimize the impact of irrelevant factors and noise, we select the top five most relevant health indicators for each battery based on their absolute Pearson correlation coefficients. These selected features serve as the input to the neural network, ensuring that the model is provided with the most informative and clean data.

3. The model addresses the challenges posed by real-world noise and battery capacity regeneration by incorporating a denoising step within the code. This step enhances the robustness of the model, making it more resilient to noisy input data and better suited for practical deployment in various battery management systems.

The following are the arrangements for the remainder of this article: Section 2 describes the methods used, including feature extraction and the proposed network model. Section 3 validates the model's effectiveness with actual battery data, presenting experimental results and analysis. Section 4 provides the conclusions of this study.

## II. METHODOLOGY

### A. Feature extraction

The data-driven health indicators are derived from the datasets. All health indicators are sourced from NASA and CALCE datasets. These extracted indicators and their aging performance are shown in Table I.

These health indicators are multidimensional features, each with varying degrees of correlation to the SOH. Including all health indicators in the output could introduce noise from less relevant features, thereby reducing prediction accuracy. Therefore, we eliminate low-correlation indicators and select high-correlation indicators for input into the network. In this

TABLE I: Details of Health Indicators (HIs)

| Abbreviation | Explanation | Aging Behavior |
|---|---|---|
| CCT | Constant current charging time | Shortens as battery ages due to increased internal resistance causing faster voltage rise |
| CVT | Constant voltage charging time | Lengthens as battery ages, with reduced current acceptance near full charge, lowering charging efficiency |
| DT | Discharge time | Shortens as battery ages, with increased internal resistance causing faster voltage drop |
| TT | Time to reach maximum temperature | Shortens as battery ages, with increased internal resistance generating more heat, causing faster temperature rise |
| R | Internal resistance | Increases as battery ages |
| CMT | Time for constant voltage charging current to drop to 1.5A | Shortens as battery ages, with decreased capacity and increased internal resistance causing faster current drop |
| CVI mean | Mean constant voltage charging current | Decreases as battery ages, with increased internal resistance and current dropping to a lower level until fully charged |
| CVI std | Standard deviation of constant voltage charging current | Increases as battery ages, with greater fluctuation |
| CCV mean | Mean constant current charging voltage | Decreases due to increased voltage drop from higher internal resistance |
| CCV std | Standard deviation of constant current charging voltage | Increases as battery ages, with greater fluctuation |
| CDV mean | Mean constant current discharging voltage | Decreases due to increased voltage drop from higher internal resistance |
| CDV std | Standard deviation of constant current discharging voltage | Increases as battery ages, with greater fluctuation |

research, we use Pearson correlation analysis to select the indicators and choose the top five as the network input. The Pearson correlation coefficient is commonly used in the analysis of the relationship between SOH and health factors [34], and its calculation principle is shown in Eq (**??**).

$$A = \frac{\sum_{i=1}^{n}(a_i - \bar{a})(b_i - \bar{b})}{\sqrt{\sum_{i=1}^{n}(a_i - \bar{a})^2}\sqrt{\sum_{i=1}^{n}(b_i - \bar{b})^2}} \qquad (2)$$

where $a_i$ and $b_i$ represent the values of the data points, with $\bar{a}$ and $\bar{b}$ denoting their respective mean values, and $n$ being the total number of data points. The Pearson correlation coefficient $A$ is an important measure of the linear correlation between two or more variables in statistics. If the value of the correlation coefficient is closer to 1, the correlation between the two variables is very strong; that is, there is a high degree of positive correlation between the two variables. The coefficients

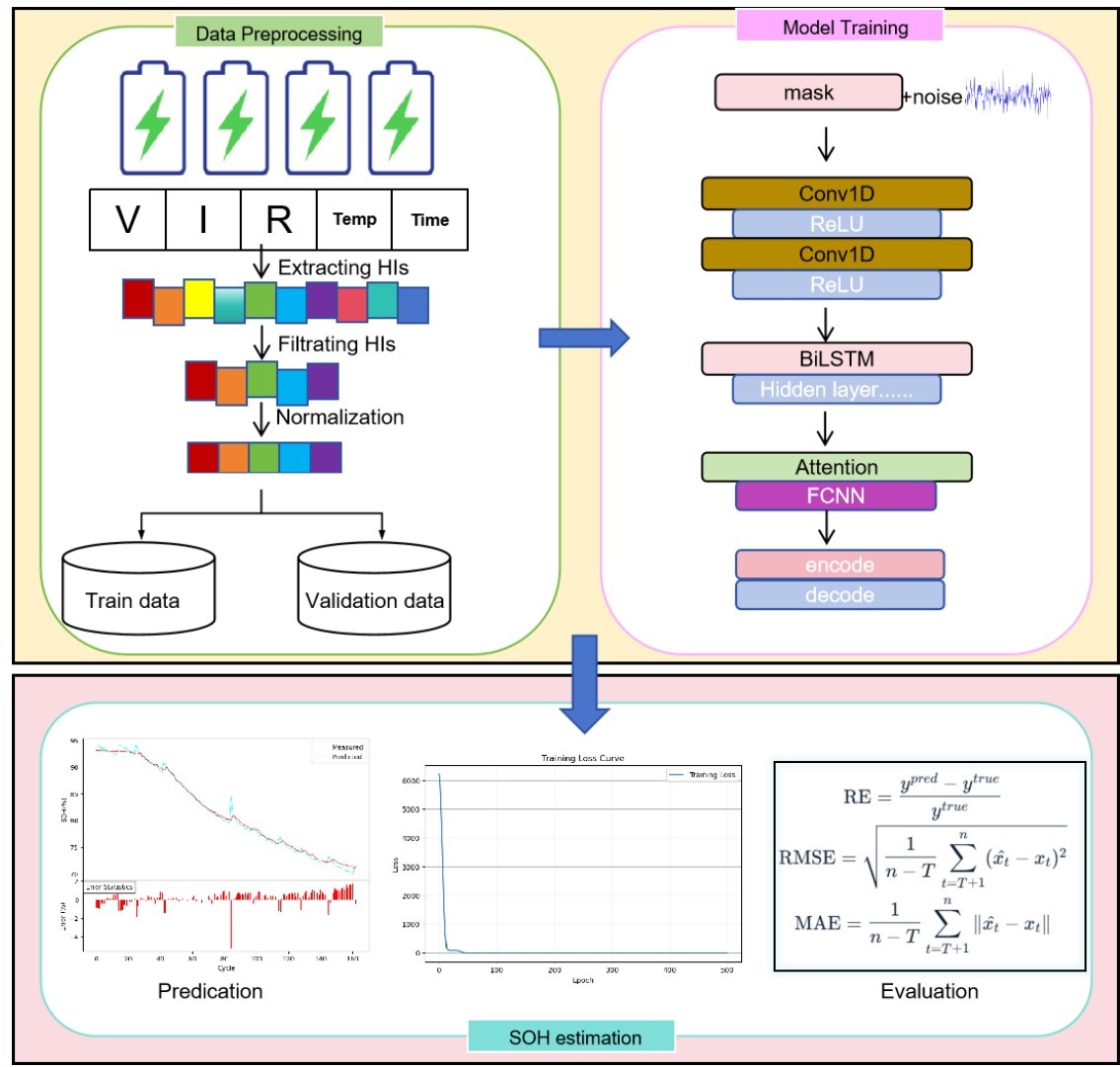

Fig. 1: Overall framwork of the proposed battery SOH estimation model.

provide an intuitive way to explain complex correlations in data and help researchers and decision-makers understand the potential relationships and impacts between the two variables.

### B. DAE-CNN-BiLSTM-Attention model

The raw input data is often noisy, especially during charge and discharge cycles. In most approaches, models input the raw data directly into the neural network without denoising, which significantly impacts prediction accuracy. The algorithm first denoises the training samples and then injects them into the deep neural network to ensure the stability and robustness of the algorithm. In this project, we propose using a denoising autoencoder (DAE) to reconstruct low-dimensional data through unsupervised learning while preserving as much information as possible [35].

In the last few years, attention mechanisms have shown great promise in various deep learning tasks [36]. In this paper, we calculate attention scores using an attention mechanism, convert them into weights with the softmax function, and then apply these weights to the outputs of the LSTM to obtain context vectors. This approach is simple to implement, computationally efficient, and well-suited for handling time-series data, highlighting important temporal information within the network.

Fig. 1. illustrates the framework of the DAE-CNN-BiLSTM-Attention model for predicting battery SOH, including denoising functionality and the CNN, BiLSTM, and Attention modules. In part A, health indicators are extracted, and the top 5 features are selected based on their Pearson correlation coefficients. These features are then normalized, with 70% used as training data and 30% as validation data. Gaussian noise is added, followed by two CNN layers to extract local features. The BiLSTM captures long-term dependencies in the sequential data, while the attention mechanism helps the network focus on the most important parts for predicting SOH. An autoencoder is employed to denoise the data by attempting to reconstruct the original data from the noisy input, thereby enhancing the robustness of the network. The model structures

TABLE II: Neural network structure and parameters.

| Model | Structure | Number of Sampling Points |
|---|---|---|
| CNN | noisy input$\rightarrow X$
Conv1D(Channel: 64/Kernel: 3)$\rightarrow$ReLU$\rightarrow$
Conv1D(Channel: 128/Kernel: 3)$\rightarrow$ReLU$\rightarrow$ | X
64
128 |
| BiLSTM | Number of bidirectional layers: 1
Hidden_size: 100 $\rightarrow$ Hidden_size * 2 | 128
200 |
| Attention | Hidden_size * 2 $\rightarrow$
Attention_size: 20 $\rightarrow$
Fc(200$\rightarrow$1) | 20
200
1 |
| Encoder | encoder_fc1: input_size * sequence_length
$\rightarrow$ hidden_size
decoder_fc2: $\rightarrow$ input_size * sequence_length | 100
X
X |

TABLE III: Experimental conditions for NASA dataset.

| | Battery | B5 | B6 | B7 | B18 |
|---|---|---|---|---|---|
| | Normial capacity(Ah) | 2 | 2 | 2 | 2 |
| | Data length | 168 | 168 | 168 | 133 |
| | Ambient temperature(°C) | 24 | 24 | 24 | 24 |
| Charge | CC(A) | 1.5 | 1.5 | 1.5 | 1.5 |
| | cut-off current(mA) | 20 | 20 | 20 | 20 |
| | CV(V) | 4.2 | 4.2 | 4.2 | 4.2 |
| Discharge | CD(A) | 2 | 2 | 2 | 2 |
| | cut-off voltage(V) | 2.7 | 2.5 | 2.2 | 2.5 |

are summarized in Table II. The loss function converges to zero, and the model's performance is quantified using RMSE and MAE metrics.

## III. EXPERIMENT RESULTS AND ANALYSIS

### A. Datasets

The data from the NASA repository was collected by the NASA Ames Prognostics Center of Excellence (PCoE) on the NASA prognostics tested [11]. NASA batteries were used to validate the proposed method [35].This study utilizes batteries B0005, B0006, B0007,and B0018,abbreviations are used in the table, as B5,B6,B7,B18 respectively,Table III shows the experimental conditions for these batteries.Fig. 2 illustrates the capacity degradation process of the NASA battery dataset.These batteries have a failure threshold of 1.4 Ah.

The CALCE dataset is a battery cycling test dataset from the Center for Advanced Life Cycle Engineering (CALCE) at the University of Maryland. CALCE batteries are widely used in battery state estimation studies and were used to validate the proposed method in [33]. This study uses batteries CS2_35, CS2_36, CS_37,and CS2_38,abbreviations are used in the table, as C35,C36,C37,C38 respectively,Table IV shows the experimental conditions for these batteries.Fig. 3 illustrates the capacity degradation process of the CALCE battery dataset.These batteries have a failure threshold of 0.77 Ah.

### B. Feature Selection

The health factors from the NASA and CALCE datasets discussed in this paper were extracted based on the capacity

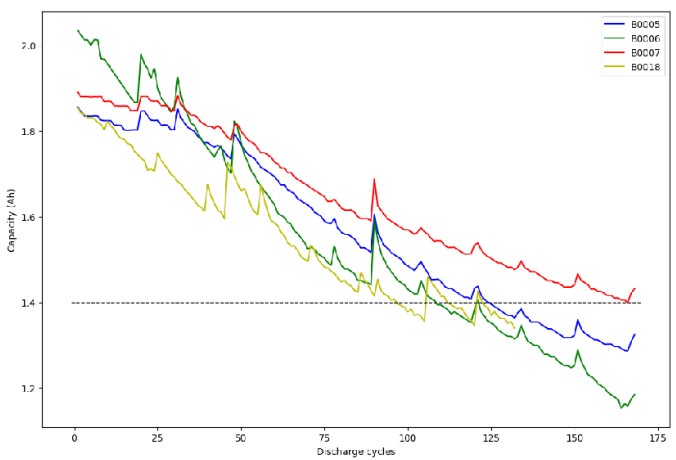

Fig. 2: NASA dataset capacity degration at ambient temperature of 24°C.

TABLE IV: Experimental conditions for calce dataset

| | Battery | C35 | C36 | C37 | C38 |
|---|---|---|---|---|---|
| | Normial capacity(Ah) | 1.1 | 1.1 | 1.1 | 1.1 |
| | Data length | 882 | 936 | 969 | 996 |
| | Ambient temperature(°C) | 1 | 1 | 1 | 1 |
| Charge | CC(A) | 0.5 | 0.5 | 0.5 | 0.5 |
| | cut-off current(mA) | 20 | 20 | 20 | 20 |
| | CV(V) | 4.2 | 4.2 | 4.2 | 4.2 |
| Discharge | CD(A) | 1 | 1 | 1 | 1 |
| | cut-off voltage(V) | 2.7 | 2.7 | 2.7 | 2.7 |

degradation characteristics shown in Fig. 2 and Fig. 3, as well as Table I in Section 2A. The relationship between the health factors and SOH was analyzed through correlation analysis using Eq (2).

As shown in table V, The top five health factors were highlighted in red ,and then these selected factors were used as inputs to the model. For instance, the inputs selected for B0005 are 'CCT', 'DT', 'TT', 'CMT','CDV mean'. Fig. 4 , Fig. 5 illustrate the correlations between the various health indicators respectively,with positive correlation in red and negative correlation in blue.In Fig. 4(a),the mean value of the constant current discharge voltage showed the strongest positive correlation, with a coefficient of 0.98.In Fig. 5(a),the

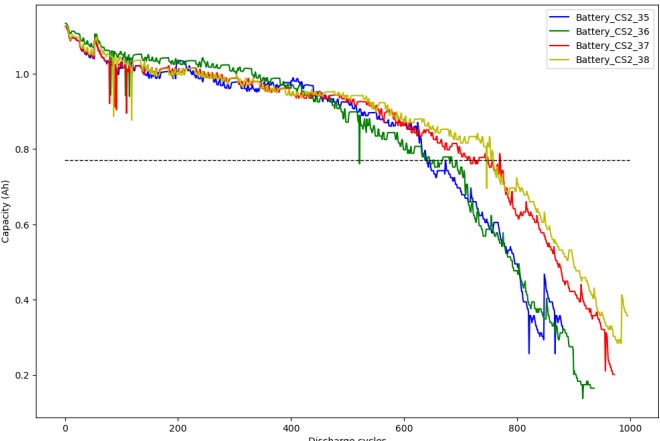

Fig. 3: CALCE dataset capacity degration at ambient temperature of 1°C.

mean value of the constant current discharge voltage and discharge time exhibited the strongest positive correlation, with a coefficient of 0.99, while internal resistance showed the strongest negative correlation, with a coefficient of -0.97.

### C. Overall performance

This study employs three commonly used metrics to quantify the performance of the model in predicting battery health status: Mean Absolute Error (MAE), Root Mean Squared Error (RMSE). The definitions of these metrics are as follows:

$$\text{MAE} = \frac{1}{n-T} \sum_{t=T+1}^{n} \|\hat{x}_t - x_t\| \tag{3}$$

$$\text{RMSE} = \sqrt{\frac{1}{n-T} \sum_{t=T+1}^{n} (\hat{x}_t - x_t)^2} \tag{4}$$

Where $C_n$ represents the length of the sequence, and $C_T$ represents the length of the training sequence samples. MAE (Mean Absolute Error) is a metric that calculates the average absolute error between measured and predicted values, which measures the average difference between them. RMSE (Root Mean Square Error) is the mean difference between the predicted value and actual values, providing the standard deviation of the errors.

To thoroughly validate the performance of the proposed DAE-CNN-BiLSTM-Attention model, we designed and conducted several experiments across multiple battery datasets. The results of these experiments are presented in Table VI, where the best-performing results are highlighted in bold. These results demonstrate the model's efficacy in predicting the State of Health (SOH) of batteries with high accuracy.

For the NASA dataset, the evaluation results show that the B0005 battery achieved a Mean Absolute Error (MAE) of 0.5075 and a Root Mean Square Error (RMSE) of 0.7064. The B0006 battery reported an MAE of 0.8462 and an RMSE of 1.2405. The B0007 battery yielded an MAE of 0.4407 and

an RMSE of 0.6337, while the B0018 battery had an MAE of 0.7258 and an RMSE of 0.9738. For the CALCE dataset, the CS2_35 battery achieved an MAE of 0.0154 and an RMSE of 0.0191, the CS2_36 battery had an MAE of 0.0266 and an RMSE of 0.0303, the CS2_37 battery obtained an MAE of 0.0207 and an RMSE of 0.0335, and the CS2_38 battery recorded an MAE of 0.0286 and an RMSE of 0.0509. Notably, except for the RMSE of CS2_36, which is slightly lower than the 0.0230 achieved by the CNN-BiLSTM-At model, and the MAE of CS2_38, which is lower than the 0.0227 achieved by the CNN-BiLSTM-At model, our proposed model outperformed other models across all other metrics, indicating its robustness and accuracy.

To further assess the contribution of the denoising step, we compared the results with and without the denoising function. The comparison highlights that while the denoising function has improved the model's performance, there remains potential for further enhancement. The DAE-CNN-BiLSTM-Attention model consistently demonstrated the lowest MAE and RMSE across the datasets, underscoring its superior predictive performance. The best evaluation result for this model was achieved on the CS2_35 battery, where it recorded a Mean Absolute Error (MAE) of 0.0154 and a Root Mean Square Error (RMSE) of 0.0191. Compared to the model without denoising, the performance improvements in MAE and RMSE were 55.4% and 3.14%, respectively, showcasing the effectiveness of the denoising step in refining the model's predictions.

Moreover, the simplicity and efficiency of the proposed model are worth noting. The model requires only one minute to complete 500 training iterations, which is significantly faster compared to the 90 minutes and 10 minutes reported in a prior study [11]. This efficiency, coupled with the model's accuracy, makes it a practical solution for real-time battery health monitoring systems.

Figures 6 and 7 provide a visual representation of the prediction results and the associated errors for the NASA and CALCE datasets, respectively. Figure 6 clearly shows that the predicted SOH values for the NASA batteries closely match the actual battery health, with all prediction errors remaining within 5%, even at peak anomaly points. This indicates the model's ability to accurately track the degradation process despite the presence of anomalies. Similarly, Figure 7 depicts the prediction and error trends for the CALCE battery health state. In this study, 70% of the battery data was used for training, and the model was tasked with predicting the entire degradation process.

The CALCE dataset presents a more challenging prediction scenario compared to the NASA dataset due to its significantly larger data volume and the presence of more anomalous noise. Despite these challenges, the model's predictions remain close to the actual degradation curve, demonstrating its robustness and ability to generalize across different datasets. While the prediction error increased slightly for the CALCE dataset, this increase is minimal, and the results still exhibit a high degree of accuracy, further validating the model's effectiveness in

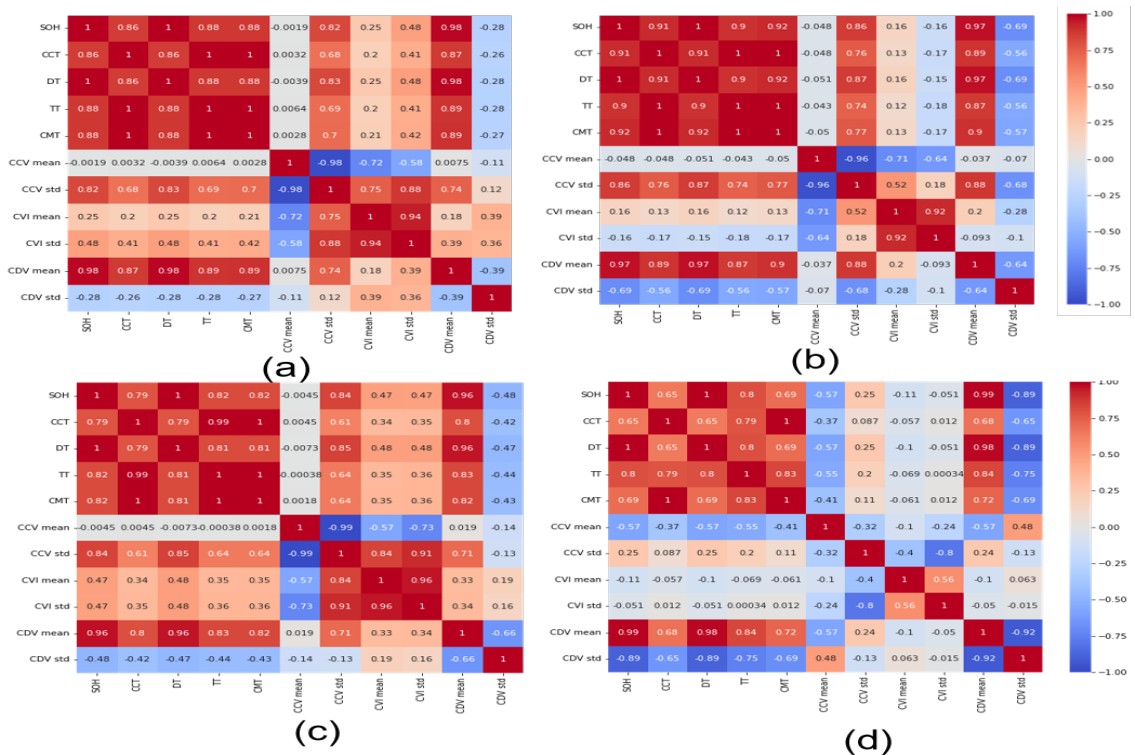

Fig. 4: NASA Pearson Correlation Heatmap:(a)B5;(b)B6;(c)B7;(d)B18.

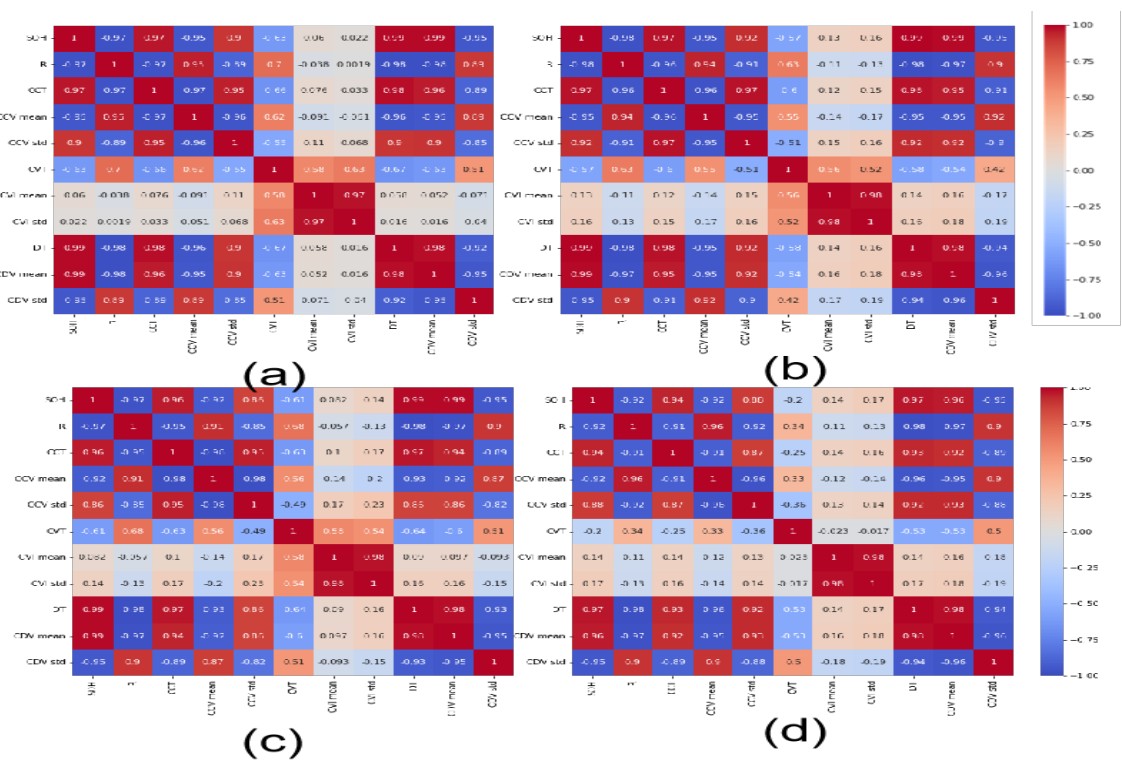

Fig. 5: CALCE Pearson Correlation Heatmap:(a)C35;(b)C36;(c)C37;(d)C38.

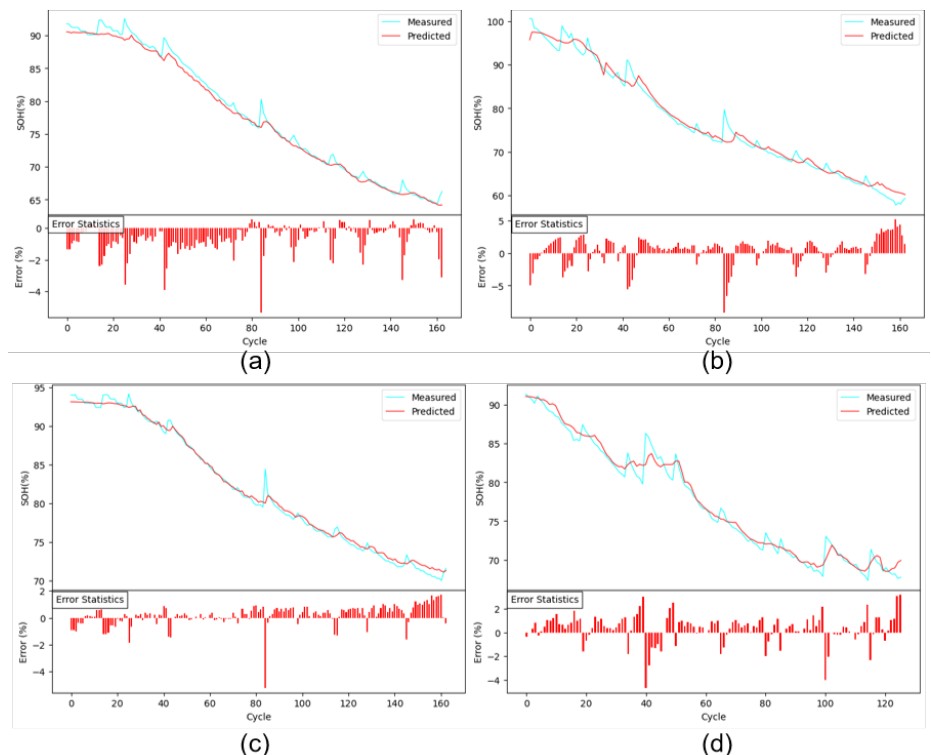

Fig. 6: NASA capacity estimation results and errors:(a)B5;(b)B6;(c)B7;(d)B18.

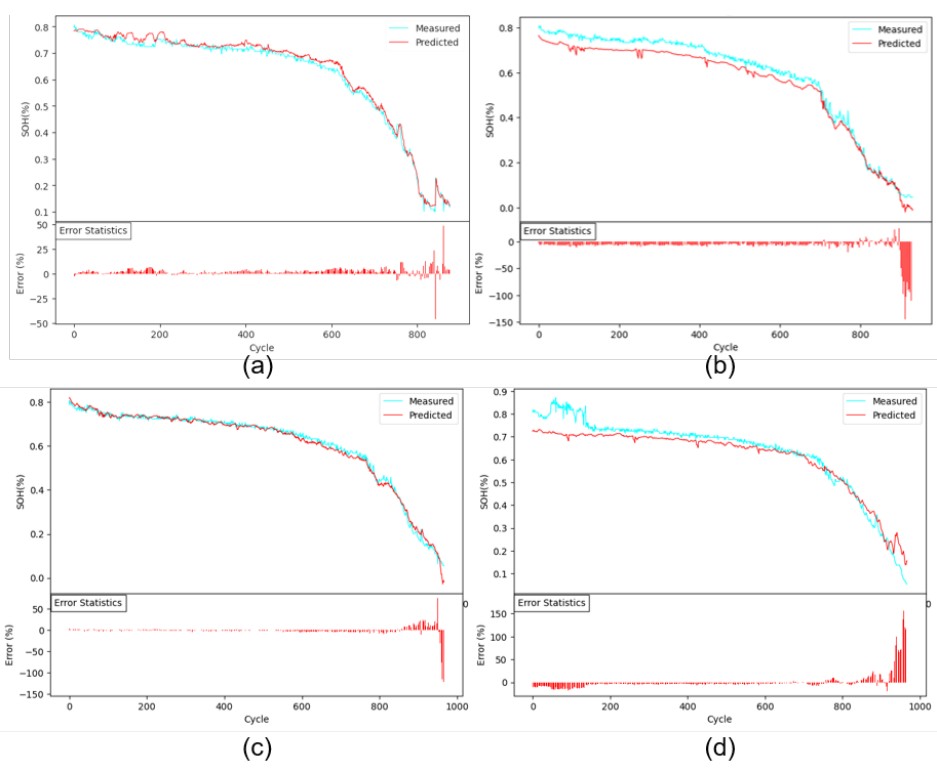

Fig. 7: CALCE capacity estimation results and errors:(a)C35;(b)CS36;(c)C37;(d)C38.

TABLE V: Correlation Coefficients of Health Indicators for Different Batteries.

| B0005 | B0006 | B0007 | B0018 |
|---|---|---|---|
| CCT 0.862586 | CCT 0.908604 | CCT 0.786684 | CCT 0.648654 |
| DT 0.999947 | DT 0.999915 | DT 0.999725 | DT 0.999773 |
| TT 0.877558 | TT 0.897890 | TT 0.816621 | TT 0.804534 |
| CMT 0.881882 | CMT 0.919521 | CMT 0.815183 | CMT 0.694932 |
| CCV mean -0.001942 | CCV mean -0.047989 | CCV mean -0.004465 | CCV mean -0.570699 |
| CCV std 0.822106 | CCV std 0.862932 | CCV std 0.839493 | CCV std 0.251537 |
| CVI mean 0.245934 | CVI mean 0.160218 | CVI mean 0.471227 | CVI mean -0.106422 |
| CVI std 0.475350 | CVI std -0.156089 | CVI std 0.468511 | CVI std -0.051191 |
| CDV mean 0.982357 | CDV mean 0.965189 | CDV mean 0.961071 | CDV mean 0.985401 |
| CDV std -0.283559 | CDV std -0.694572 | CDV std -0.482625 | CDV std -0.892615 |

| CS2_35 | CS2_36 | CS2_37 | CS2_38 |
|---|---|---|---|
| R -0.969031 | R -0.975631 | R -0.968516 | R -0.922052 |
| CCT 0.967323 | CCT 0.969335 | CCT 0.955910 | CCT 0.942224 |
| CCV mean -0.952510 | CCV mean -0.951288 | CCV mean -0.922861 | CCV mean -0.921456 |
| CCV std 0.897771 | CCV std 0.917804 | CCV std 0.855091 | CCV std 0.875672 |
| CVT -0.626522 | CVT -0.565320 | CVT -0.612713 | CVT -0.197397 |
| CVI mean 0.060142 | CVI mean 0.133853 | CVI mean 0.081966 | CVI mean 0.142492 |
| CVI std 0.022159 | CVI std 0.156692 | CVI std 0.144904 | CVI std 0.173008 |
| DT 0.991876 | DT 0.994180 | DT 0.991499 | DT 0.967231 |
| CDV mean 0.990909 | CDV mean 0.989263 | CDV mean 0.988342 | CDV mean 0.955991 |
| CDV std -0.947330 | CDV std -0.949600 | CDV std -0.945006 | CDV std -0.952266 |

TABLE VI: SOH estimate MAEs and RMSEs on NASA and CALCE datasets.

| Datasets | Metrics | LSTM | At-LSTM | CNN-BiLSTM | CNN-BiLSTM-At | DAE-CNN-BiLSTM-At |
|---|---|---|---|---|---|---|
| B0005 | MAE | 1.0882 | 1.0880 | 0.8882 | 0.5521 | **0.5075** |
|  | RMSE | 1.5428 | 1.3567 | 1.3393 | 0.7334 | **0.7064** |
| B0006 | MAE | 1.6684 | 1.2518 | 1.1133 | 1.2459 | **0.8462** |
|  | RMSE | 2.2141 | 1.7345 | 1.6578 | 1.2806 | **1.2405** |
| B0007 | MAE | 1.1695 | 1.1872 | 0.9839 | 0.5992 | **0.4407** |
|  | RMSE | 1.3116 | 1.5503 | 1.4796 | 0.9050 | **0.6337** |
| B0018 | MAE | 1.4277 | 1.2273 | 0.9233 | 0.7266 | **0.7258** |
|  | RMSE | 1.8202 | 1.8140 | 1.2973 | 1.1352 | **0.9738** |
| CS2_35 | MAE | 0.0488 | 0.0470 | 0.0485 | 0.0478 | **0.0154** |
|  | RMSE | 0.0267 | 0.0294 | 0.0228 | 0.0197 | **0.0191** |
| CS2_36 | MAE | 0.0373 | 0.0382 | 0.0341 | 0.0337 | **0.0266** |
|  | RMSE | 0.0391 | 0.0341 | 0.2546 | **0.0230** | 0.0303 |
| CS2_37 | MAE | 0.0315 | 0.0226 | 0.0335 | 0.0371 | **0.0207** |
|  | RMSE | 0.0571 | 0.0262 | 0.0380 | 0.0364 | **0.0335** |
| CS2_38 | MAE | 0.0384 | 0.0358 | 0.0261 | **0.0227** | 0.0286 |
|  | RMSE | 0.0511 | 0.0498 | 0.0522 | 0.0713 | **0.0509** |

real-world applications.

## IV. CONCLUSION

Accurately estimating the State of Health (SOH) of batteries is critical for effective battery management, and establishing a reliable prediction network is key. We have proposed a data-driven hybrid neural network for SOH prediction. Initially, we have extracted over ten features from the batteries and selected the top five features based on their absolute Pearson correlation coefficients for input into the network. The Convolutional Neural Network (CNN) has been employed to extract features from the noisy input data, followed by the Bidirectional Long Short-Term Memory (BiLSTM) network, which has learned the degradation information of the battery. An Attention mechanism has been applied to focus on important information, and finally, the autoencoder-decoder has restored the noisy data to its original state, thereby enhancing the model's adaptability and stability. The proposed model has been validated on different battery datasets and has demonstrated lower Mean Absolute Error (MAE) and Root Mean Square Error (RMSE) compared to other models.

In the future, we plan to make improvements in the following three directions:

1.Further enhancement of the denoising function to improve

performance.

2.Increased focus on the spatial relationships between features.

3.A comparative analysis with more single classical models to identify their strengths and weaknesses for improving prediction accuracy.

ACKNOWLEDGMENT

The authors would like to thank Teacher Huang for his encouragement and support.

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
