# OpenReview forum: "A Data-Driven DAE-CNN-BiLSTM-Attention Prediction Model for the State of Health of Lithium-ion Batteries"
_IEEE.org/ICIST/2024/Conference — IEEE ICIST 2024 Conference Submission_

### Official Review · Reviewer_zHwu · 2024-08-26
**Review Comments for Manuscript No. 254**

**Rating:** 7
**Confidence:** 4

**Review:**

1. To enhance the readability of the manuscript, the following issues need to be addressed: First, there are many abbreviations used in the paper, but some of them are not spelled out when they first appear, while others are explained multiple times. Additionally, the reference formatting is inconsistent throughout the manuscript. Lastly, there are noticeable errors in the citation of equations and figures, with some figures being incorrectly referenced. The authors should carefully review and correct these issues.

2. In some tables, the lack of separation lines makes it difficult to distinguish the corresponding text and meaning for each row, as seen in Table II. Furthermore, does Table III imply that all experimental conditions are the same? If so, is this assumption too idealized? The authors are advised to revise these details carefully.

3. In Table VI, the RMSE of At-LSTM on the CS2 37 and CS2 38 datasets is better than that of DAE-CNN-BiLSTM-At. The authors need to provide a specific explanation for this and ensure that the highlighted data is accurate.

4. In Figures 6 and 7, could additional details be included to better reflect the length of the training process?

5. Similar to studies like [24], the authors should consider providing an algorithm or outline in the form of pseudocode to describe the implementation of the entire model.

6. One of the innovations of this paper is a complex hybrid model. While this combination sounds promising, it requires further discussion on why this specific architecture was chosen. Additionally, the authors emphasize the role of DAE in noise suppression and use Pearson correlation analysis to achieve this. However, Pearson correlation can only capture the strength of linear relationships. How do the authors plan to address the nonlinear aspects in future work?

---

### Official Review · Reviewer_3j1g · 2024-08-27
**A Data-Driven DAE-CNN-BiLSTM-Attention Prediction Model for the State of Health of Lithium-ion Batteries**

**Rating:** 7
**Confidence:** 4

**Review:**

In this paper, the authors extracted over ten health indicators and designed a hybrid model: DAE-CNN-BiLSTM-Attention. This work is well organized and novel. Below are some comments. (1) The contributions should be illustrated in a clearer manner. For example, what is the main improvement of the paper compared to the existing results. The authors should explain the unique contributions of this paper. (2) The simulation results should be explained more carefully. (3) The paper is well presented, spelled correctly. I recommend authors to carefully read the entire paper to find possible misspellings.

---

### Official Review · Reviewer_PtAt · 2024-08-27
**Good paper, but there are some caveats.**

**Rating:** 7
**Confidence:** 4

**Review:**

The article introduces a hybrid neural network model called DAE-CNN-BiLSTM-Attention for predicting the State of Health (SOH) of lithium-ion batteries. This model integrates Denoising Autoencoders (DAE), Convolutional Neural Networks (CNN), Bidirectional Long Short-Term Memory (BiLSTM) networks, and an Attention mechanism to improve prediction accuracy, especially in noisy real-world data. The model is validated on NASA and CALCE datasets, showing superior performance compared to existing methods.

Comments:

1.Provide more details on the rationale behind selecting exactly five health indicators and whether different numbers of features were tested.

2.Elaborate on how the denoising process enhances model performance, possibly with visual examples.

3.Include comparisons with simpler or classical models to highlight the improvements made by the proposed approach.

4.Discuss the model's limitations.

---

### Decision · Program_Chairs · 2024-09-08

Accept (Oral)